# Optimizing Robotic Task Sequencing and Trajectory Planning on the Basis of Deep Reinforcement Learning

**DOI:** 10.3390/biomimetics9010010

**Published:** 2023-12-27

**Authors:** Xiaoting Dong, Guangxi Wan, Peng Zeng, Chunhe Song, Shijie Cui

**Affiliations:** 1State Key Laboratory of Robotics, Shenyang Institute of Automation, Chinese Academy of Sciences, Shenyang 110016, Chinasongchunhe@sia.cn (C.S.); cuishijie@sia.cn (S.C.); 2Key Laboratory of Networked Control Systems, Shenyang Institute of Automation, Chinese Academy of Sciences, Shenyang 110016, China; 3Institutes for Robotics and Intelligent Manufacturing, Chinese Academy of Sciences, Shenyang 110169, China; 4University of Chinese Academy of Sciences, Beijing 100049, China

**Keywords:** robot task sequencing, trajectory planning, co-optimization, deep reinforcement learning, robotic manufacturing

## Abstract

The robot task sequencing problem and trajectory planning problem are two important issues in the robotic optimization domain and are solved sequentially in two separate levels in traditional studies. This paradigm disregards the potential synergistic impact between the two problems, resulting in a local optimum solution. To address this problem, this paper formulates a co-optimization model that integrates the task sequencing problem and trajectory planning problem into a holistic problem, abbreviated as the robot TSTP problem. To solve the TSTP problem, we model the optimization process as a Markov decision process and propose a deep reinforcement learning (DRL)-based method to facilitate problem solving. To validate the proposed approach, multiple test cases are used to verify the feasibility of the TSTP model and the solving capability of the DRL method. The real-world experimental results demonstrate that the DRL method can achieve a 30.54% energy savings compared to the traditional evolution algorithm, and the computational time required by the proposed DRL method is much shorter than those of the evolutionary algorithms. In addition, when adopting the TSTP model, a 18.22% energy reduction can be achieved compared to using the sequential optimization model.

## 1. Introduction

An increasing number of robots are being deployed on production floors to achieve intelligent manufacturing. This has motivated the development of production optimization technology in robotics. Robot task sequencing [1] and trajectory planning [2] are two traditionally separate optimization problems in robotics. The task sequencing problem focuses on finding the optimal task execution sequence for the robot in order to achieve certain objectives. It is analogous to the traveling salesman problem (TSP), but it is more complex because of the robot’s kinematic redundancy (also known as the IK solution). The robot trajectory planning problem involves determining the timing of the motion law that the robot follows along a predefined geometric path while satisfying specific requirements, such as trajectory smoothness and accuracy, and achieving desired objectives, such as those related to the execution time, energy consumption, vibration, and their combinations; a general overview is given in [3].

In the robotic work cell, there exists a unique group of application scenarios that involve task sequencing and trajectory planning problems simultaneously, such as spot welding, freeform surface inspection, or spray painting [4]. In these applications, the robot is required to visit a set of task points with no predefined sequence to perform corresponding tasks, and it must finally return to its initial state. Clearly, the visiting sequence of the task points and the trajectories followed for the robot to reach each task point have strong effects on the robot’s production efficiency. Additionally, different task-point visiting sequences can lead to alterations in the robot’s moving path, consequently impacting the robot’s movement trajectories. Therefore, it is necessary to address the task sequencing and trajectory planning problems simultaneously.

However, the robot task sequencing and trajectory planning problems are traditionally treated as two separate issues. The robot motion trajectories are often predefined at the production control level, and then, the production scheduling level conducts the task sequence planning. This sequential optimization model diminishes the optimization space of the production problems and can improve productivity to a certain extent, but the optimization solutions are often suboptimal because of the neglect of the underlying synergies associated with the optimization objectives among the task sequencing and trajectory planning problems. In addition, the solution space for the TSTP problem grows exponentially with the number of manufacturing points. For an exhaustive search, the computational complexity is $O(m!g^{m})$, where *m* is the number of task points and *g* is the number of IK solutions for each point. Traditional methods adopted to solve task sequencing and trajectory planning problems, such as metaheuristics and dispatching rules, face challenges in terms of time efficiency and ensuring high-quality solutions. These methods either suffer from slow processing times or struggle to consistently achieve optimal solutions that meet various objectives. Over the past few years, deep reinforcement learning algorithms (DRL) [5] have enabled significant advancements; their excellent self-learning and self-optimization qualities enable them to solve complex decision-making problems quickly and accurately, which has resulted in their extensive utilization across various robot work cell optimization problems [6].

Motivated by the aforementioned co-optimization requirements and DRL algorithms, in this paper, we take the optimization problems in task sequencing and trajectory planning as a monolithic problem and model it as a Markov decision process. Furthermore, a DRL-based optimization method is developed to address the above co-optimization problem. The major contributions are as follows.

To the best of our knowledge, we are the first to combine the traditionally separate optimization problems in task sequencing and trajectory planning into a monolithic problem, called the robot TSTP problem, providing an integrated view in the discrete manufacturing domain.To solve the above TSTP problem, we employ a DRL-based policy for decision optimization. During the learning process, a specific state representation, action space, and reward function are carefully designed. Typically, in the action space, each action step considers the selection of task points, IK solutions, and trajectory parameters concurrently.Given the absence of a benchmark test for scheduling and control co-optimization, the feasibility of the proposed TSTP model and the effectiveness of the DRL are validated through a demonstration abstracted from a specific real-world case: a spot-welding task in an automation plant with a UR5 robot.

The rest of this article is organized as follows. Section 2 provides a summary of the related work. Section 3 formulates the mathematical model of the robot TSTP problem under study. Section 4 presents the methodology. Section 5 presents the experiments and results. Section 6 concludes this study.

## 2. Related Work

This paper describes a comprehensive study that considers the combined optimization of the robot task sequencing and trajectory planning problems. Because of the scarcity of work in this field, this section discusses the research statuses of the task sequencing and trajectory planning problems individually.

### 2.1. Existing Works for Robot Task Sequencing Problem

The robot task sequencing problem is essentially a tour planning problem and can be viewed as a variation of the traveling salesman problem or one of its extensions [7,8]. The formulation of the robot task sequencing problem as a mixed-integer nonlinear program problem is a common approach to its solution. However, searching for exact solutions via the mixed-integer nonlinear program method can be computationally intensive, especially for larger instances. To tackle the large-scale robot task sequencing problem, heuristics are applied, such as GA [9], PSO [10], etc. The work presented in [11] was one of the first to use GA to address the robot task sequencing problem; the authors designed an innovative encoding method that contained both the task sequence and corresponding configurations’ information. In their experiments, the computational time for a 6-DoF robot with a problem size of 50 points was approximately 1800 s. Xidias et al. [12] and Zacharia et al. [13] extended this study by considering obstacles in 2D and 3D environments; they used the bump surface concept to capture obstacle information and employed a genetic algorithm to attain global solutions. However, their experiments were limited to small problem sizes, with only up to 15 target points being considered.

To reduce the computational time required for the large-scale robot task sequencing problem, the clustering algorithm and its improvement are regarded as a promising solution. Gueta et al. [14] and Wong C et al. [15] attempted to solve the robot task sequencing problem via clustering. They addressed the robot task sequencing problem in the same way, firstly dividing the task points into a predetermined number of clusters based on their topological locations. Subsequently, the robot task sequencing problem was transformed into the problem of finding a tour across clusters and determining the subsequent visiting sequences for the navigation of points within each cluster. The difference was that the former study was conducted in a joint space, while the latter was in a Cartesian space. Encouragingly, for a 6-DoF robot with one thousand task points, the clustering algorithm was able to find the solution within 200 s.

More recently, the RoboTSP algorithm [16] has emerged as another method that could compete with clustering in terms of speed. It was able to produce a high-quality solution in less than a minute for a complex drilling task involving 245 targets, with an average of 28.5 configurations per target. Hiu-Hung et al. [1] optimized the robot task sequencing problem using a two-tier model that only required seconds to obtain solutions for hundreds of points. To begin, they employed a real-coded twin-space crowding evolutionary algorithm to establish the robot configuration. Then, they optimized the task point sequence of the robot through a heuristic bidirectional reference mechanism. Despite the effectiveness of these heuristics in dealing with the robot task sequencing problem, the authors separated the optimization of task sequencing and trajectory planning. Consequently, this setup caused the performance of the downstream sequential planning to be constantly restricted by the trajectory planning results. Since robot task sequencing and trajectory planning are closely intertwined, it is imperative to optimize them concurrently in order to meet the requirements on both fronts.

### 2.2. Existing Works for Robot Trajectory Planning

Trajectory optimization is an important tool for the minimization of industrial robot operation times and energy consumption, which is a relevant research issue for many scholars.

Trajectory optimization is an important tool for the minimization of industrial robot operation times and energy consumption, which is a relevant research issue for many scholars. Time optimality is the earliest and most commonly used trajectory optimization index. Bobrow JE et al. [17] were the first to solve the minimum-time robot trajectory planning problem using the conventional linear feedback control theory. Lee Y D et al. [18] then attempted to use GA to solve the time-optimal trajectory planning for a two-link manipulator. With the development of intelligent algorithms, some algorithms with good performance have been introduced to address the robot trajectory planning problem, such as the DE algorithm [19], the PSO algorithm [20,21], the ant colony algorithm [22], the whale algorithm [23], and so on. However, focusing solely on minimizing time as the objective in robot trajectory planning frequently results in significant vibrations of the robot, leading to a loss of tracking and positioning accuracy and even exacerbating motor wear. Therefore, the co-optimization of time optimality and trajectory smoothness becomes an important research area. In the work [24], a teaching–learning-based optimization algorithm was proposed to solve the optimal time–jerk trajectory planning for a 6-DoF welding robot. Zhang T et al. [25] presented a practical time-optimal and smooth trajectory planning algorithm and applied it to robot arm trajectory planning successfully. Works [26,27,28] adopted the NSGA-II algorithm, new convex optimization approach, and triple nurbs curves with bidirectional interpolation algorithm, respectively, to solve the optimal time–jerk trajectory planning problems for 6-DoF industrial manipulators, which greatly enriched the optimization algorithm library for robot trajectory planning.

In recent years, there has been growing concern regarding energy-efficient robot trajectory planning and the simultaneous minimization of energy and time, driven by the increasing costs of energy [29]. Various practical methods of optimizing energy along specified paths in robotic systems have been developed, such as scaling the reference trajectory, as shown in works [30,31]. Lennartson B. et al. [32] extended the time-scaling process by introducing a dynamic scaling factor to save more energy. Zhou J et al. [2] proposed a modified algorithm for B-spline feedrate curves with a callback mechanism, resulting in energy and time savings of over 45% compared to the conservative feed in the case of sculptured surface machining under complex constraints. Hou R et al. [33] formulated a time–energy optimization model using a phase plane based on the Riemann approximation method and utilized an iterative learning algorithm with neural networks to achieve an optimal trajectory for the precise control of industrial robots. W Sang et al. [34] utilized the particle swarm optimization algorithm to derive the best trajectory parameters to achieve the minimization of time and energy. Shi Xiang Dong et al. [35] focused on the dynamic optimal trajectory planning for a robot with a time–energy–jerk co-optimization objective and employed a fast and elitist genetic algorithm and multi-objective particle swarm optimization algorithm to address it.

## 3. Problem Formulation

The study described in this paper is regarding the simultaneous optimization of the robot task sequencing problem and trajectory planning problem. To apply optimization methods, the mathematical model of the robot TSTP problem is formulated. To enhance the understanding of the TSTP model, the mathematical models of the robot task sequencing and the trajectory planning problems are briefly introduced in advance.

### 3.1. The Mathematical Model of the Robot Task Sequencing Problem

For an *n*-DoF robot that is arranged to execute a manufacturing task with *m* task points $P=P_{1},P_{2},\ldots ,P_{m}$, the moving cost of the robot from task point *l* to *h* is denoted as $C_{l,h}$, and the robot task sequencing optimization objective can be expressed as
(1)$$\begin{array}{c} min\;\sum_{l=1}^{m}\sum_{h=1,h\neq l}^{m}C_{l,h}x_{l,h}\\ subject\;to\;\forall l,\sum_{l=1,l\neq h}^{m}x_{l,h}=1,\\ \forall h,\sum_{h=1,h\neq l}^{m}x_{l,h}=1\\ \forall l,\forall h,y_{l}-y_{h}+mx_{l,h}=m-1\\ l\neq 1,\;h\neq 1,\;l\neq h,x_{l,h}\in 0,1\\ And\;\forall l,\;y_{l}=0,\;y_{l}\in I \end{array}$$
where the binary variable $x_{l,h}$ is assigned a value of 1 when the path from point *l* to *h* is chosen; otherwise, it is assigned a value of 0. In addition, the dummy variables $y_{l}$ and $y_{h}$ are employed to ensure that all points are covered once within the selected sequence. It should be noted that in the robot TSTP problem, $C_{l,h}$ is not a constant. Rather, $C_{l,h}$ is a variable to be evaluated according to the trajectory curve after the *l*th and *h*th points are configured.

### 3.2. Joint Configuration Selection for the Task Points

The multiplicity of the IK solutions is one of the typical characteristics of the robot kinematics model, which means that the robot can reach the same task point in many different joint configurations; see Figure 1a. This type of multiplicity makes the robot task sequencing problem more complex but also increases the optimization potential of the robotic work cell. If a specific joint configuration is selected for the robot to reach a task point, then the task point is configured; see Figure 1b.

Assume that the robot needs to move from task point $P_{1}\left(x_{1},y_{1},z_{1},Rx_{1},Ry_{1},Rz_{1}\right)$ to $P_{2}\left(x_{2},y_{2},z_{2},Rx_{2},Ry_{2},Rz_{2}\right)$. According to the robot inverse kinematics model $q=\Gamma^{-1}\left(P\right)$, two sets of IK solutions, $Sol_{1}=\left\{q_{1},q_{2},\ldots ,q_{g}\right\}$, $Sol_{2}=\left\{q_{1}^{\prime },q_{2}^{\prime },\ldots ,q_{g}^{\prime }\right\}$, can be obtained, where *g* is the number of IK solutions and $q_{i}=\left\{\theta_{1},\theta_{2},\ldots ,\theta_{n}\right\}$ is a group of joint configurations of the robot that can reach the task point $P_{i},i\in 1,2$. The robot transformation cost from joint configuration *k* to *r* is denoted as $t_{k,r}$, and the selection of the joint configurations for the task points can be described as follows: select a joint configuration $q_{k}$ from the IK solution set $Sol_{1}$ and select another joint configuration $q_{r}$ from the IK solution set $Sol_{2}$ to ensure that the $t_{k,r}$ is minimum.

### 3.3. The Mathematical Model of the Robot Trajectory Planning Problem

The primary objective of optimal trajectory planning in this study is to minimize the value of the robot’s moving time and energy consumption simultaneously. Assume that the robot needs to move from pose $q_{1}=\left\{\theta_{1},\theta_{2},\ldots ,\theta_{n}\right\}$ to pose $q_{2}=\left\{\theta_{1}^{\prime },\theta_{2}^{\prime },\ldots ,\theta_{n}^{\prime }\right\}$, with the constraints of ${\dot{q}}_{1}=0$, and ${\ddot{q}}_{1}=0$, ${\dot{q}}_{2}=0$ and ${\ddot{q}}_{2}=0$. According to the robot dynamics model, the joint torque can be calculated by
(2)$$\begin{array}{c} \tau =M\left(q\right)\ddot{q}+C\left(q,\dot{q}\right)\dot{q}+F\left(\dot{q}\right)+G\left(q\right) \end{array}$$
where q, $\dot{q}$, and $\ddot{q}$ are the vectors of the joint angles, joint velocities, and joint accelerations, respectively. M(q) is the mass matrix, $C(q,\dot{q})$ is the Coriolis and centripetal coupling matrix, $F(\dot{q})$ is the friction force vector (comprising both viscous friction and Coulomb friction), G(q) is the gravity vector, and τ is the joint torque vector. Thus, the energy consumption can be articulated as
(3)$$\begin{array}{c} E=\int_{0}^{T}|\tau \cdot \dot{q}|dt\\ subject\;to\;\tau_{j}^{min}\leq |\tau_{j}|\leq \tau_{j}^{max}\\ {\dot{q}}_{j}^{min}\leq |{\dot{q}}_{j}|\leq {\dot{q}}_{j}^{max} \end{array}$$
where $\tau_{j}^{min}$ and $\tau_{j}^{max}$ and ${\dot{q}}_{j}^{min}$ and ${\dot{q}}_{j}^{max}$ are the lower and upper bounds of the torque and velocity of the *j*th joint, respectively. Then, a model for optimal time–energy trajectory planning is developed within the joint space, described as
(4)$$\begin{array}{c} min\;\alpha_{1}T+\alpha_{2}E \end{array}$$
where *T* is the moving time for the robot from pose $q_{1}$ to $q_{2}$. $\left\{\alpha_{1},\alpha_{2}\right\}\in R^{+}$ ensures that *T* and *E* are in the same order of magnitude. Note that, in the robot TSTP problem, the moving time *T* is a variable that is determined by the IK solution selection and trajectory curve quality.

### 3.4. The Mathematical Model of the Robot TSTP Problem

The above analysis shows that the robot moving cost is closely related to the moving trajectories, which are further influenced by the task point execution sequence and joint configuration selection when a specific task is executed. Traditionally, the robot trajectory planning problem is addressed at the production control level, while the robot task sequencing problem is addressed at the scheduling level. This type of sequential optimization model ignores the underlying synergies associated with the optimization objectives among the task sequencing and trajectory planning problems and cannot obtain truly global optimal solutions. To cope with this problem, we construct a co-optimization model that integrates the robot task sequencing and trajectory planning problems as a holistic problem, called the robot TSTP model.

The robot TSTP problem can be described as follows. An *n*-DoF robot manipulator is arranged to execute a manufacturing task with *m* task points $P=\{P_{1},P_{2},\ldots ,P_{m}\}$ with no predefined order. Each task point has six attributes $P_{i}=\left(x_{i},y_{i},z_{i},Rx_{i},Ry_{i},Rz_{i}\right)$ to define its position and orientation. The robot TSTP problem seeks to simultaneously plan a set of trajectories $traj_{opt}=\left(traj_{0},traj_{1},\ldots ,traj_{m-1}\right)$ and find a task point execute sequence $P_{opt}=\left\{P_{x},P_{y},\ldots ,P_{m},\ldots ,P_{k}\right\}$ to achieve time and energy consumption minimization; see Figure 2. This can be formulated as
(5)$$\begin{array}{c} min\;C_{P_{0},P_{x}}+C_{P_{x},P_{y}}+\ldots +C_{P_{i},P_{i+1}}+\ldots +C_{P_{k},P_{0}} \end{array}$$
(6)$$\begin{array}{c} C_{P_{i},P_{i+1}}=\alpha_{1}T_{i}+\alpha_{2}E_{i} \end{array}$$
where $C_{P_{i},P_{i+1}}$ represents the moving cost of the robot from task point $P_{i}$ to $P_{i+1}$ and $T_{i}$ and $E_{i}$ are the time and energy consumption correspondingly. $P_{0}$ is the robot’s initial state. $\{\alpha_{1}$, $\alpha_{2}\}\in R^{+}$ ensure that $T_{i}$ and $E_{i}$ are in the same order of magnitude. It should be noted that $T_{i}$ and $E_{i}$ are variables evaluated according to the trajectory $traj_{i}$ between task points $P_{i}$ and $P_{i+1}$. Assuming that the robot moves from $P_{i}$ using the *l*th configuration to $P_{i+1}$ using the *r*th configuration, the smallest moving time $T_{i,min}$ can be calculated by the following:(7)$$\begin{array}{c} T_{i,min}=max\left(\frac{|\theta_{j}^{i+1,r}-\theta_{j}^{i,l}|}{{\dot{\theta }}_{jmax}}\right),j=0,1,\ldots ,n \end{array}$$

In Equation (Equation 7), the robot is always required to move at its maximum speed. Although this can reduce the robot’s moving time significantly, it can also lead to increased wear and tear on the robot’s components and higher energy consumption. To balance the robot’s moving speed and energy consumption, a set of time factors ω={1.0,1.05,1.1,1.15,1.2} is designed to make the robot moving time $T_{i}$ variable and controllable. In this case, the robot moving time $T_{i}$ can be calculated by the following:(8)$$\begin{array}{c} T_{i}=T_{i,min}*\omega_{i} \end{array}$$
where $\omega_{i}\in \omega$, which is also the control parameter of the trajectory planning that should be optimized together with the robot task points sequence in our Co-RTSATP problem.
(9)$$\begin{array}{c} \tau_{i}=M\left(q_{i}\right){\ddot{q}}_{i}+C\left(q_{i},{\dot{q}}_{i}\right){\dot{q}}_{i}+F\left({\dot{q}}_{i}\right)+G\left(q_{i}\right) \end{array}$$
(10)$$\begin{array}{c} E_{i}=\int_{T_{0}}^{T_{i}}|\tau_{i}\cdot {\dot{q}}_{i}|dt\\ subject\;to\;\tau_{j}^{min}\leq |\tau_{i,j}|\leq \tau_{j}^{max}\\ {\dot{q}}_{j}^{min}\leq |{\dot{q}}_{i,j}|\leq {\dot{q}}_{j}^{max} \end{array}$$
$\tau_{i}$ is a vector of the joint torque; it is a variable that varies from $\tau_{P_{i}}$ to $\tau_{P_{i+1}}$ along with time. $\tau_{i,j}$ and $q_{i,j}$ are the torque and velocity, respectively, of the *j*th joint when moving from task point $P_{i}$ to $P_{i+1}$.

### 3.5. Assumptions

In developing the methodology to optimize the robot TSTP problem, the following requirements must be satisfied:The robot needs to pass through each task point only once and return to its initial pose after completing all task points;No obstacles should exist between the task points;After reaching a task point, the robot should move directly to the next task point; the time taken to finish the task is not considered when calculating the cycle time because the time required to finish a task depends only on a specific process;The robot kinematics and dynamics should be known.

## 4. Methodology

In this section, the robot trajectories between task points are planned using the quintic polynomial interpolation (QPI) algorithm, and the global optimization process of the robot TSTP problem is implemented via the DRL algorithm.

### 4.1. The QPI Algorithm

The QPI algorithm is widely used in the realm of robot trajectory planning because of the smoothness and lack of acceleration jumps. In the robot TSTP problem, the general mathematical expression of the QPI algorithm is
(11)$$\left\{\begin{array}{cc} q\left(t\right)=a_{1}+a_{1}t+a_{2}t^{2}+a_{3}t^{3}+a_{4}t^{4}+a_{5}t^{5} & \\ \dot{q}\left(t\right)=a_{1}+2a_{2}t+3a_{3}t^{2}+4a_{4}t^{3}+5a_{5}t^{4} & \\ \ddot{q}\left(t\right)=2a_{2}+6a_{3}t+12a_{4}t^{2}+20a_{5}t^{3} \end{array}\right.$$
where $a_{0},a_{1},a_{2},a_{3},a_{4},a_{5}$ satisfy:(12)$$\left[\begin{array}{c} a_{0}\\ a_{1}\\ a_{2}\\ a_{3}\\ a_{4}\\ a_{5} \end{array}\right]={\left[\begin{array}{cccccc} 1 & t_{0} & t_{0}^{2} & t_{0}^{3} & t_{0}^{4} & t_{0}^{5}\\ 0 & 1 & 2t_{0} & 3t_{0}^{2} & 4t_{0}^{3} & 5t_{0}^{4}\\ 0 & 0 & 2 & 6t_{0} & 12t_{0}^{2} & 20t_{0}^{3}\\ 1 & t_{f} & t_{f}^{2} & t_{f}^{3} & t_{f}^{4} & t_{f}^{5}\\ 0 & 1 & 2t_{f} & 3t_{f}^{2} & 4t_{f}^{3} & 5t_{f}^{4}\\ 0 & 0 & 2 & 6t_{f} & 12t_{f}^{2} & 20t_{f}^{3} \end{array}\right]}^{-1}\left[\begin{array}{c} q(0)\\ \dot{q}\left(0\right)\\ \ddot{q}\left(0\right)\\ q(f)\\ \dot{q}\left(f\right)\\ \ddot{q}\left(f\right) \end{array}\right]$$

According to Equation (Equation 11), the quintic polynomial coefficient matrix $A={\left[a_{0},\ldots ,a_{5}\right]}^{T}$ can be calculated when the joints’ angle positions, velocities, and accelerations are determined and the time *t* is given. In the robot TSTP problem, the joints’ angles, velocities, and accelerations at the start and end points are known, as shown in Formula (12), and $T_{i}$ is obtained by the DQN. Thus, A can be solved.
(13)$$\left\{\begin{array}{cc} t_{0}=0,t_{f}=T_{i} & \\ q\left(t_{0}\right)=q_{i},q\left(t_{f}\right)=q_{i+1} & \\ \dot{q}\left(t_{0}\right)=0,\dot{q}\left(t_{f}\right)=0 & \\ \ddot{q}\left(t_{0}\right)=0,\ddot{q}\left(t_{f}\right)=0 \end{array}\right.$$

An example of using QPI to plan the robot moving trajectories is shown in Figure 3, where $q_{j}\left(0\right)=1.81\;$ rad, $q_{j}\left(2.3\right)=4.62\;$ rad, ${\dot{q}}_{j}\left(0\right)=0\;$ rad/s, ${\dot{q}}_{j}\left(2.3\right)=0\;$ rad/s, ${\ddot{q}}_{j}\left(0\right)=0\;{\mathrm{rad}/s}^{2}$, and ${\ddot{q}}_{j}\left(2.3\right)=0\;{\mathrm{rad}/s}^{2}$. The curves of the joint angle, angular velocity, and angular acceleration are smooth, with no mechanical vibration, which is beneficial for the robot’s service life.

### 4.2. DRL Algorithm

The deep Q-learning neural network algorithm (DQN) is a type of DRL algorithm that integrates the RL techniques and supervised learning [36]. The proposed DQN framework for the robot TSTP problem is illustrated in Figure 4.

The DQN algorithm utilizes two deep neural networks: the evaluation network *Q* and the target network $\hat{Q}$. First, the agent observes the initial state $s_{0}$ of the robot TSTP problem environment and computes the state features’ values. Then, the agent takes an action to interact with the environment using the ϵ−Greedy policy. Next, the agent receives a reward from the environment, and the *Q*-network is trained through gradient descent based on the calculated error. Finally, a new state $s^{\prime }$ and its corresponding state features’ values are computed before the next decision point. In this way, the agent constantly interacts with the environment and records its experiences $\left(s,a,r,s^{\prime }\right)$ within the environment. The reward (*r*) received for the action (*a*) in a given state (*s*) and the resulting new state ($s^{\prime }$) after the action are stored in the experience replay memory. When a sufficient number of experiences has been accumulated, an amount equal to the size of the mini-batch is randomly selected from the experience replay memory to train the Q-network. As the learning progresses, the experience is constantly updated. By reusing and learning from past experiences, the agent can learn from a more diverse and robust set of experiences, which, in turn, helps to decorrelate the data samples and break the temporal correlations in the sequence of experiences. This is important to prevent the learning process from falling into local optima and to make the learning process more stable and data-efficient.

#### 4.2.1. Solution Representation

In the DQN algorithm, the solution of the robot TSTP problem can be expressed by the task points sequence vector, joint configuration number vector, and time factor vector; each vector contains *m* elements. In the task points sequence vector $\left\{\lambda_{1},\lambda_{2},\ldots ,\lambda_{m}\right\}$, element $\lambda_{i}\in \left[1,m\right]$ is described by a task point number. In the joint configuration vector $\left\{c_{1},c_{2},\ldots ,c_{m}\right\}$, element $c_{i}\in \left[1,g\right]$ is represented by an IK solution number. In the time factor vector $\left\{\omega_{1},\omega_{2},\ldots ,\omega_{m}\right\}$, element $\omega_{i}\in \left\{1.00,1.05,1.10,1.15,1.20\right\}$. A solution representation is illustrated in Figure 5, where the red parts connected by the green arrows indicate that the robot reaches task point 9 with the 7th joint configuration, the moving time from the last task point 5 to point 9 is equal to $T_{5,min}\times 1.05$, and the same applies for other task points. Note that the last element marked in blue represents the time factor taken by the robot to move from the initial pose to the first task point 3, as well as the time factor from the last task point 6 back to the initial pose.

#### 4.2.2. Decoding

The fitness of the robot TSTP problem can be calculated when a corresponding solution is provided. An example of the encoding process for the solution from Figure 5 is explained as follows. The robot first selects task point 3 for processing. According to the pose information $P_{3}=\left(x_{3},y_{3},z_{3},Rx_{3},Ry_{3},Rz_{3}\right)$ and the inverse kinematic model $q=\Gamma^{-1}\left(P\right)$, a set of IK solutions $Sol=\{q_{1},q_{2},\ldots ,q_{g}\}$ is formed. Each IK solution $q_{i}$ represents a type of joint configuration used to reach task point 3, and the 6th joint configuration $s_{6}$ is chosen for the robot to reach task point 3 according to the joint configuration number vector. Because this is the first task point, the robot needs to move from the initial pose $q_{0}$ to $q_{3}^{6}$ by the time $max(\frac{|\theta_{j}^{9,7}-\theta_{j}^{5,7}|}{{\dot{\theta }}_{jmax}})\times 1.05$ according to the time factor vector. Next, we plan the moving trajectory $traj_{0⟶3}$ using the QPI algorithm between the initial pose $q_{0}$ and pose $q_{3}^{6}$. Once the trajectory expression is known, the moving time $T_{0\to 3}$ and energy consumption $E_{0\to 3}$ can be calculated. By analogy, after all the task points in the solution are calculated, the total time and energy consumption of the robot in executing the manufacturing task can be obtained by summation.

#### 4.2.3. State Features

Good state features should strike a balance between capturing environmental information accurately and avoiding unnecessary computational overhead. Based on the manufacturing task characteristics and optimization objective, four state features are designed as follows.
(1)Task state TS(t)


(14)
$$\begin{array}{c} TS\left(t\right)=(I_{1}\left(t\right),I_{2}\left(t\right),\ldots ,I_{m}\left(t\right) \end{array}$$


TS(t) is a binary vector whose length is equal to the task points number *m*, and $I_{i}\left(t\right)=1$ indicates that task point $p_{i}$ has been executed at decision point *t*; otherwise, it has not.
(2)Robot joint state JS(t)


(15)
$$\begin{array}{c} JS\left(t\right)=(\theta_{1}\left(t\right),\theta_{2}\left(t\right),\ldots ,\theta_{n}\left(t\right)) \end{array}$$


JS(t) is an angle vector whose length is equal to the robot joint number *n*, and $\theta_{i}\left(t\right)$ is the current angle of the *i*th joint. For example, $\theta_{2}\left(t\right)=2.5$ rad indicates that the angle of the 2nd joint is 2.5 rad at decision point *t*.
(3)Current total moving time CT(t)


(16)
$$\begin{array}{c} CT\left(t\right)=\Sigma_{i=1}^{\kappa }T_{i} \end{array}$$


CT(t) is a cumulative variable that is equal to the sum of the moving time spent by the robot at decision point *t*. κ represents the last completed task point $P_{\kappa }$ at decision point *t*.
(4)Current total energy consumption CE(t)
(17)$$\begin{array}{c} CE\left(t\right)=\Sigma_{i=1}^{\kappa }E_{i} \end{array}$$

CE(t) is a cumulative variable that is equal to the sum of energy consumed by the robot at decision point *t*. κ represents the last executed task point $P_{\kappa }$ at decision point *t*.

#### 4.2.4. Actions

Practically, the agent has three decision variables at each decision point *t*. They are as follows:The task point number $\lambda_{i}$ to be executed next;The joints configuration number $c_{i}$ for the robot to reach the task point $p_{\lambda_{i}}$;The time factor $\omega_{i}$ for the robot to move from the current pose to task point $p_{\lambda_{i}}$.

Thus, an action can be expressed by a decision vector $a\left(t\right)=\left(\lambda_{i},c_{i},\omega_{i}\right)$, where $\lambda_{i}\in \left[1,m\right]$, $c_{i}\in \left[1,g\right]$ and $\omega_{i}\in \left\{1.00,1.05,1.10,1.15,1.20\right\}$. Algorithm 1 illustrates the procedure for the implementation of actions at each decision point.
**Algorithm 1** Action implementation1:in put:probabilityϵ,alloutputnodesQ−values∈DQN2:$\mathrm{out}\;\mathrm{put}:an\;action\;a=(\lambda_{i},c_{i},\omega_{i})$3:generatearandomnumberξ4:ifξ≤ϵ,then5:   selecttheactionwhoseQ−valueisthelargest6:else7:   selectanactionrandomly8:end

#### 4.2.5. Reward Function

After action $a_{t+1}=\left(\lambda_{i}^{\prime },c_{i}^{\prime },\omega_{i}^{\prime }\right)$ is implemented, the agent can observe at least one of the following three types of environment state features and obtain a corresponding reward. The specific reward calculation procedure is illustrated in Algorithm 2.

The task point $p_{\lambda_{i}^{\prime }}$ has been completed, and the agent receives a substantial negative reward.The agent completes its target, $TS_{t+1}=TS_{end}$, and the agent receives a substantial positive reward.The agent completes the task point $p_{\lambda_{i}^{\prime }}$ and obtains a normal reward that relates to the time and energy increment.

**Algorithm 2** Reward definition
1:

in put:CTandCEatdecisionpointtandt+1

2:

$\;\;\;\;\;\;TS\left(t\right)=(I_{1}\left(t\right),I_{2}\left(t\right),\ldots ,I_{m}\left(t\right))\;at\;decision\;point\;t$

3:

$\;\;\;\;\;\;TS\left(t+1\right)=(I_{1}^{\prime }\left(t+1\right),I_{2}^{\prime }\left(t+1\right),\ldots ,I_{m}^{\prime }\left(t+1\right))\;at\;decision\;point\;t+1$

4:

$\;\;\;\;\;\;action\;\;a_{t+1}=\left(\lambda_{i}^{\prime },c_{i}^{\prime },\omega_{i}^{\prime }\right)at\;\;decision\;point\;t+1$

5:

$\mathrm{out}\;\mathrm{put}:\;reward_{r+1}$

6:

$\mathrm{if}\;I{\left(t+1\right)}_{\lambda_{i}^{\prime }}=1$

7:

$\;\;\;reward_{r+1}=-300$

8:

$\mathrm{else}\;\mathrm{if}\;TS\left(t+1\right)=TS_{end}$

9:

$\;\;\;reward_{r+1}=10$

10:

else

11:

ΔCT=CT(t+1)−CT(t)

12:

ΔCE=CE(t+1)−CE(t)

13:

$\;\;\;reward_{r+1}=-\left(w_{1}\Delta CT+w_{2}\Delta CE\right)$

14:

end




#### 4.2.6. DQN Topology and Training

The DQN algorithm proposed in this study consists of one input layer, three fully connected hidden layers, and one output layer. The number of nodes in the input and output layers matches the state feature and action numbers. Rectified linear unit (ReLU) activation functions are used in the hidden layers. The online network $Q_{e}$ is trained through the stochastic gradient method *RMSprop* according to the error $Error=\frac{1}{\left|A\right|}\sum_{i=1}^{\left|A\right|}{\left({\hat{Q}}_{i}-Qi\right)}^{2}$, while the target network $\hat{Q}$ is updated by $\hat{Q}=Q$ every five steps.

## 5. Experiments and Results

Robots can be programmed to perform repetitive and time-consuming tasks; when the number of task points is large, even small improvements in the production system can yield substantial benefits. Therefore, solving the robot TSTP problem for actual robotic work cells is very valuable.

The proposed approach was tested with four sets of experiments. The experimental data were randomly generated based on a spot-welding task in a real automation plant with a UR5 robot. Actually, any six-axis articulated robot is suitable for this research. All experiments were implemented in Python 3.7 on an Intel Core i7 3.0-GHz computer with 10 GB RAM and Windows 10 OS.

### 5.1. Robot Model

The UR5, which is the specific robot used for the experiments, is a 6-DoF robot that only has rotational joints. The standard D-H parameters derived from the Python robotics toolbox are presented in Table 1. Since the UR5 is manipulated by intersecting its last three axes at a point, only eight IK solutions are available for each task point, i.e., $Sol=\{q_{1},q_{2},\ldots ,q_{8}\}$.

### 5.2. Parameter Setting

In the DQN algorithm, three key parameters, ϵ in ϵ-greedy policy, reward decay γ, and learning rate *l*, are calibrated. The levels of the three parameters are ϵ∈{0.80,0.85,0.90,0.95}, γ∈{0.5,0.7,0.9}, and l∈{0.001,0.005,0.01,0.015,0.02}.The three parameters have 4∗3∗5=60 combinations. Calibration experiments are conducted on the instance of 10 task points with all combinations. The result in each combination is the average fitness in 10 repeats. Figure 6 illustrates the factors’ level trends of the above three parameters. According to Figure 6, we set ε=0.9,γ=0.9,l=0.005, and the other parameters are as follows: episode number L=2000, memory size M=300, and mini batch size $min_{batch}=64$. In addition, parameters $\alpha_{1}$ and $\alpha_{2}$ in optimization objective Equation (Equation 6) are set to 0.2 and 0.01, respectively, based on the calibration experiment results.

### 5.3. DQN Performance Testing

With the above parameter settings, we test the global optimization performance and convergence of the DQN algorithm based on the instance of 10 task points. In this case, the comprehensive time–energy consumption cost is 118.43, in which the cycle time and energy consumption are 18.38 s and 351.87 J, respectively. When solving the robot TSTP problem, the DQN algorithm shows an excellent global optimization ability; it is always able to continue learning during the decision process. As shown in Figure 7a, in the first 50 episodes, the agent constantly interacts with the environment to accumulate experience and optimize its decisions, and its decision-making improves rapidly. Then, the accumulated experience of interacting with the environment gradually becomes saturated, the decision-making ability improves steadily, and it finally finds the optimal solution at 1209 episodes. With the progression of the learning process, the loss function value gradually decreases to zero; see Figure 7b. This shows that the DQN algorithm has good convergence when solving the robot TSTP problem.

### 5.4. DQN Algorithm Solves Large-Scale TSTP Problems

To test the DQN algorithm’s performance in solving large problems, we conducted experiments with different problem sizes. Moreover, another two groups of experiments whose optimization objectives are only time and only energy are conducted to verify the rationality of taking time–energy consumption as an optimization objective in the robot TSTP problem. Each experiment was conducted 10 times, and the average results were collected, as shown in Figure 8. It is obvious that the DQN algorithm could solve the robot TSTP problem with different sizes efficiently. When taking the time–energy consumption as the optimization objective, the agents could always find a compromise solution in which the cycle time and energy consumption were both slightly larger than those considering cycle time or energy consumption optimization only, but they could save time or energy significantly. For example, for 100 task points, the moving time obtained by the time–energy cooperation model is 5% higher than the time-optimal model, but the energy consumption is reduced by 35%. Likewise, compared with the energy-optimal model, the energy consumption obtained by the time–energy cooperation model is 4.01% higher, but the moving time is reduced by 25.47%. As is well known, production is a complex process of multi-resource integration, and production efficiency can be affected by many factors, which are always in conflict with each other, such as the robot moving time and energy consumption. Therefore, this type of compromise solution of multi-objective optimization is more closely aligned with the actual requirements of manufacturing.

### 5.5. Comparison with Sequential Optimization Model

Comparing studies that address similar problems is challenging. This complexity stems from variations in the approaches used to tackle the problem, as well as the considerations of robotic constraints and the specific industrial environment. Additionally, there is no universal reference for the robot TSTP problem (refer to [4] for benchmarks and evaluation limitations). Despite these constraints, we conducted a comparison of our TSTP model with [37] (abbreviated as Seq-opt model) by applying its principle to the same problem, as they only considered the robot task sequencing problem under the condition that the robot always moves at its maximum joint speed between all task points.

Figure 9a shows that the robot moving time obtained by the Seq-opt model is comparable to that of the TSTP model or even shorter. However, the corresponding energy consumption is much higher than that of the TSTP model; see Figure 9b. For example, in the experiment with 50 task points, the robot moving time obtained by the Seq-opt model is only 3.58% shorter than that of the TSTP model, but the energy consumption increases by 23.71%.

The relationship between a robot’s energy consumption and moving speed is complex and cannot be expressed by a simple linear model. The results of the experiments confirm that the TSTP model has the capability to effectively explore and strike a balance between the moving speed of the robot and its energy consumption. This model is able to find solutions that minimize both the moving time of the robot and its energy consumption.

### 5.6. Comparison with Other Algorithms

To investigate the performance of the DQN, we conducted comparisons with swarm intelligence evolution algorithms. The comparison algorithms include GA (see [11]) and the differential evolution (DE) algorithm (see [18]), with a population size of 100, and other parameters match the reference. Each comparison is performed 10 times, and the average results are recorded.

Figure 10a shows the time–energy cost for all experiments. Figure 10b reports the computation time required by each algorithm. For small-scale problems, the algorithms find solutions of comparable quality because of the relatively small solution space and low computational complexity. As the problem scale increases, the proposed DQN algorithm outperforms the others. In particular, when the task points m=100, the DQN algorithm can achieve an improvement of up to 28.80% compared to the GA algorithm. The DQN algorithm demonstrates a notably quicker decision process in all experiments, especially as the number of task points increases. For example, the computation time required by the proposed DQN algorithm is only a fifth of that of the DE algorithm when the number of task points m=100. This is because the DQN algorithm employs a reinforcement learning framework, which focuses on learning from past experiences and gradually improves its performance. This framework enables the algorithm to make quicker and more informed decisions, resulting in reduced computational time. On the other hand, the DE algorithm typically requires a large number of iterations and evaluations of the objective function, resulting in longer computation times.

### 5.7. Physical Experiments

To validate the robustness and practicality of the proposed TSTP model and DQN approach, we carried out two sets of experiments on a real robot, where all parameter settings matched those of the previous simulation experiment.

#### 5.7.1. The Validation of the Robot TSTP Model

In this section, two types of experiments are presented. The first one adopted the robot TSTP model, while the other one used the Seq-opt model; both took the DQN algorithm as the global optimization method and used the QPI algorithm for trajectory planning. To assess the effectiveness of the TSTP model across different complexities, tasks with 10 and 50 task points were included in the validation process; the experimental results are presented as follows.

The voltage and current values of the motor, along with their change curves, were documented during the robot’s operation. Subsequently, the power curve and energy consumption were calculated based on these data and are presented in Figure 11. When employing the TSTP model, the robot completed the task with 10 points in 12.179 s, resulting in an energy consumption of 1929.127 J. In contrast, when utilizing the Seq-opt model, the robot’s movement took 14.303 s, consuming 2110.864 J of energy. Through analysis of the power curves, it is evident that the TSTP model guided the robot to execute tasks more steadily despite a slightly longer overall time, enabling efficient energy conservation. The superiority of the TSTP model was more obvious when the number of task points increased; see Figure 12. When using the TSTP model, the robot completed the task with 50 points in 42.86 s, resulting in energy consumption of 7212.83 J. In contrast, when utilizing the Seq-opt model, the robot’s movement took 39.51 s, consuming 8819.45 J of energy. Despite the moving time being 8.47% longer than that of the Seq-opt model, the TSTP model resulted in an energy consumption that is 18.22% lower than the Seq-opt model.

#### 5.7.2. The Performance Validation of the DQN Algorithm

In this section, we conducted three groups of experiments to test the performance of the DQN algorithm in solving the robot TSTP problem. The first group of experiments used the DQN algorithm, the second group used the DE algorithm, and the third group used the GA algorithm. Each group of experiments encompassed 10 and 50 task points. The TSTP model was employed in all experiments.

The voltage and current values of the motor, along with their change curves, were documented during the robot’s operation. Subsequently, the power curve and energy consumption were calculated based on these data and are presented in Figure 13. Upon analyzing the data, it is evident that the DQN algorithm is capable of proficiently addressing the robot TSTP problem. However, when the dataset was small, its performance was comparable to that of the traditional evolutionary algorithm. As the problem size increased, the superiority of the DQN algorithm became increasingly evident. For instance, in Figure 14, the optimal solution achieved by the DQN algorithm requires 17.83% less time and 25.28% less energy consumption compared to the DE algorithm, as well as 21.63% less time and 30.54% less energy consumption than the GA algorithm.

## 6. Conclusions

This work studied the simultaneous optimization of the robot task sequencing problem and trajectory planning problem, briefly called the robot TSTP problem. The objective is the simultaneous minimization of the robot’s moving time and energy consumption. The robot TSTP problem is an NP-hard problem; it involves task point sequencing, IK solution calculation and selection, trajectory parameter determination, and trajectory planning. The solution space and computational complexity are much larger than those of any single optimization problem. To solve the robot TSTP problem, a hybrid method that combines the DQN algorithm and the QPI algorithm is proposed. First, the TSTP problem is modeled as a Markov decision process; then, we use the DQN algorithm to select the task points, IK solutions, and trajectory parameters at each decision step. Finally, the QPI algorithm is used to plan the moving trajectories between task points.

Through detailed experiments, we evaluated the competitiveness of our proposed DQN algorithm in solving the robot TSTP problem in terms of computation speed and solution quality. In addition, the experimental results confirm the superiority of the TSTP model. The TSTP model extends the optimization space and enhances the DQN algorithm, enabling it to find more excellent solutions than any of the sequential optimization models. Furthermore, the TSTP model also increases the production flexibility of the robotic cell because it can easily adjust the scheduling and controlling parameters to respond to changes in production targets or conditions. Finally, when solving the TSTP problem, setting the time–energy optimization objective is more reasonable than setting a single objective, as it can explore a solution that can improve production efficiency while maximizing energy use. Currently, our research work only considers one robot’s production time–energy optimization, which limits the application of the proposed approach. The future direction is to apply these approaches to (a) multi-robot production environments, (b) stochastic dynamic production environments, and (c) a combination of the above.

## Figures and Tables

**Figure 1 biomimetics-09-00010-f001:**
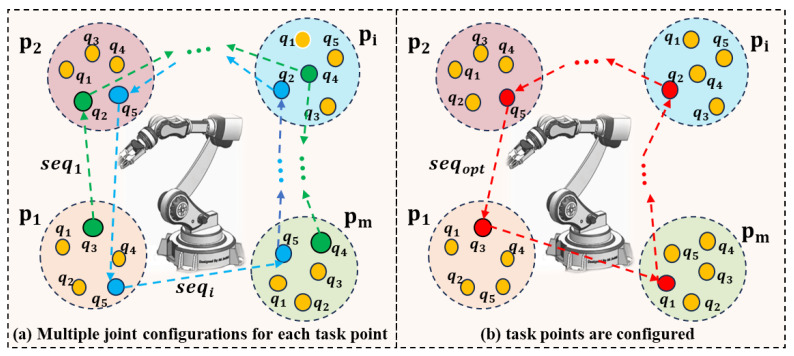
Joint configuration selection for task points. (**a**). The blue and green directed lines represent two different task points execution sequences, while the dot of the same color represent the joint configuration of the corresponding task point. (**b**). Example of an optimal task points execution sequence and corresponding joint configurations.

**Figure 2 biomimetics-09-00010-f002:**
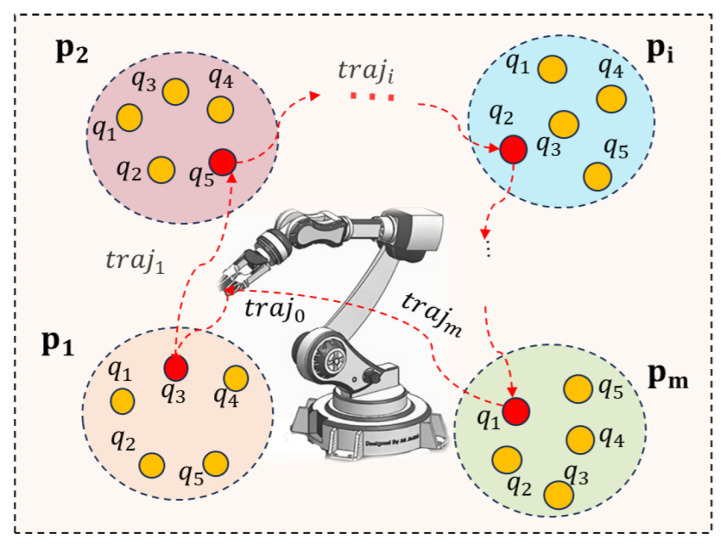
Illustration of the robot TSTP problem. The red directed curves and dots represent the optimal task points execution sequence and corresponding robot moving trajectories between task points.

**Figure 3 biomimetics-09-00010-f003:**
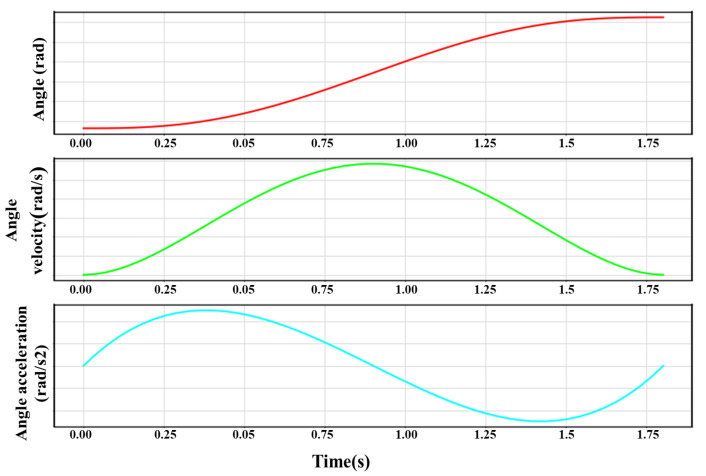
An example of robot trajectory planning by the QPI algorithm.

**Figure 4 biomimetics-09-00010-f004:**
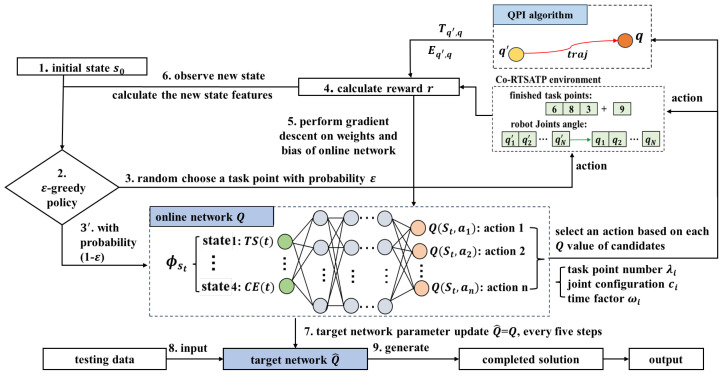
Algorithm framework for the robot TSTP problem.

**Figure 5 biomimetics-09-00010-f005:**
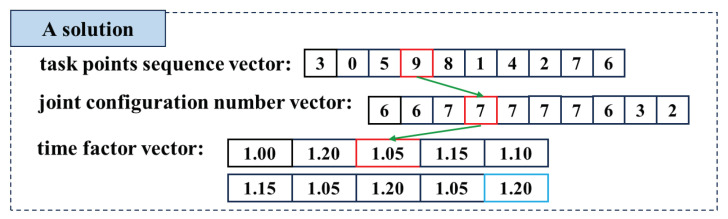
Solution representation in the DQN algorithm.

**Figure 6 biomimetics-09-00010-f006:**
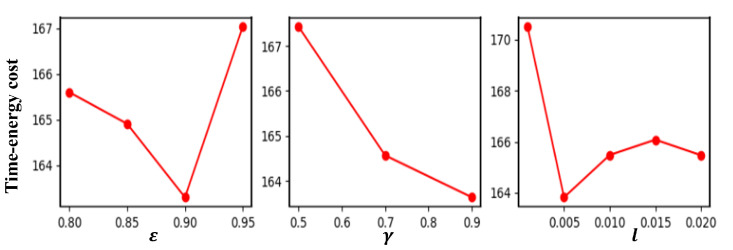
Factors’ level trends of parameters in the DQN algorithm.

**Figure 7 biomimetics-09-00010-f007:**
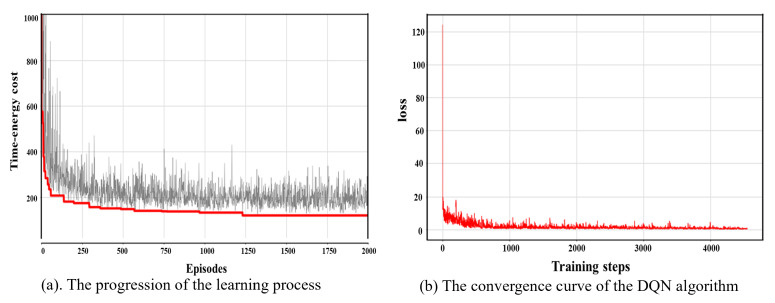
Decision-making process of the DQN algorithm when solving the Co-RTSATP problem.

**Figure 8 biomimetics-09-00010-f008:**
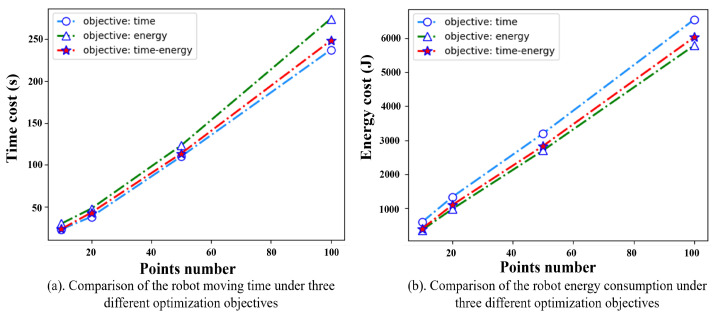
Results comparison of using DQN algorithm performance under different problem sizes.

**Figure 9 biomimetics-09-00010-f009:**
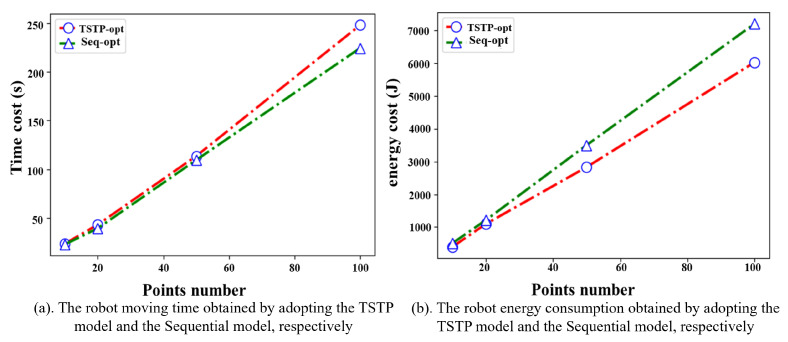
Comparison between the TSTP model and the sequential optimization model.

**Figure 10 biomimetics-09-00010-f010:**
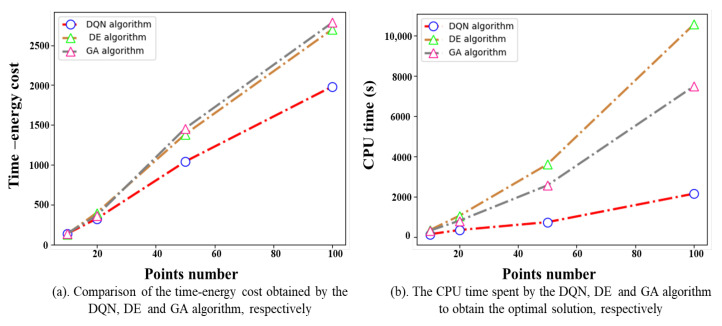
Performance comparison between the DQN algorithm and the others.

**Figure 11 biomimetics-09-00010-f011:**
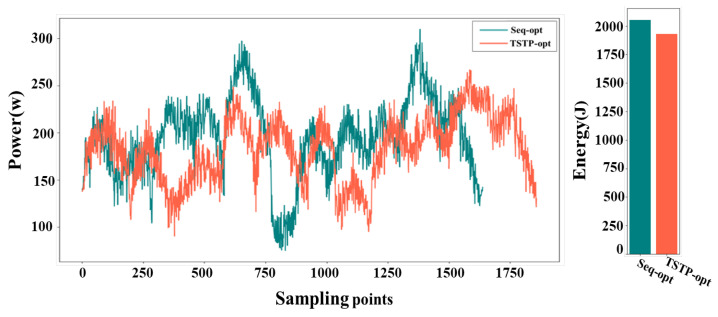
The power curves and energy consumption of the robot when executing the task involving 10 points using the TSTP model or Seq-opt model.

**Figure 12 biomimetics-09-00010-f012:**
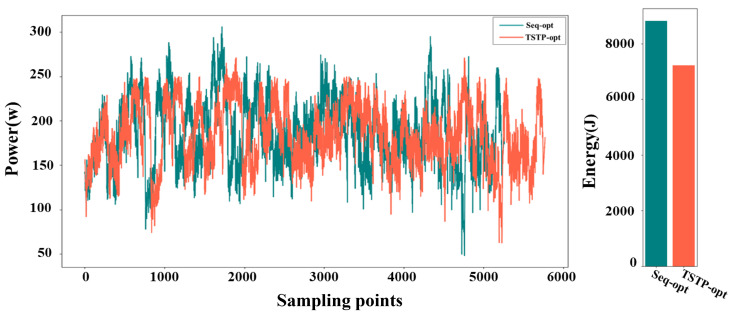
The power curves and energy consumption of the robot when executing the task of 50 points using the TSTP model or Seq-opt model.

**Figure 13 biomimetics-09-00010-f013:**
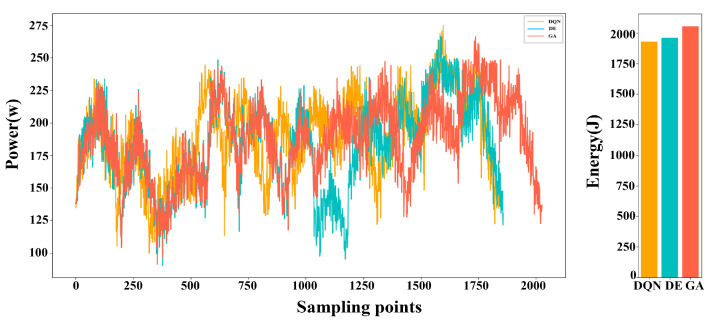
The power curves and energy consumption of the robot when executing the task of 10 points using the DQN algorithm, DE algorithm, and GA.

**Figure 14 biomimetics-09-00010-f014:**
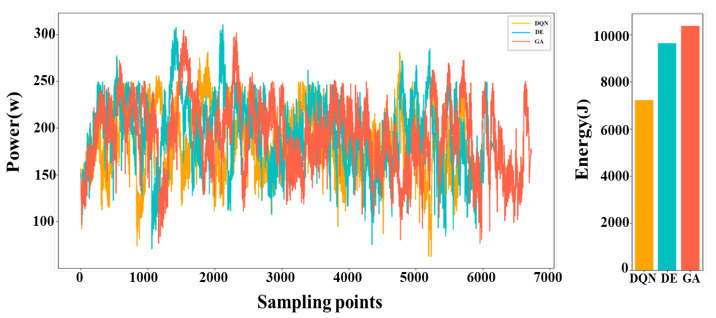
The power curves and energy consumption of the robot when executing the task of 50 points using the DQN algorithm, DE algorithm, and GA.

**Table 1 biomimetics-09-00010-t001:** Denavit–Hartenberg (DH) parameters of the UR5.

$q_{i}$	$q_{i}$	$a_{i}$	$\alpha_{i}$
$q_{1}$	0.08946	0	π
$q_{2}$	0	−0.425	0
$q_{3}$	0	−0.3922	0
$q_{4}$	0.1091	0	π
$q_{5}$	0.09465	0	−π
$q_{6}$	0.0823	0	0

## Data Availability

Data are contained within the article.

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
