# Peer review of "Optimizing Robotic Task Sequencing and Trajectory Planning on the Basis of Deep Reinforcement Learning"

_biomimetics, 2023, doi:10.3390/biomimetics9010010_

Round 1
Reviewer 1 Report
Comments and Suggestions for Authors
The paper presents an innovative approach to address the challenges in robotic optimization, mainly focusing on the Robot Task Sequencing Problem (RTSP) and the Robot Optimal Trajectory Planning Problem (ROTP). These problems are typically solved in isolation, which may lead to suboptimal solutions. This paper proposes a holistic framework, Co-RTSATP, integrating both problems and utilizing Deep Reinforcement Learning (DRL) to solve them.
Drawbacks:
1. While integrating RTSP and ROTP theoretically enhances optimization, it significantly increases the problem's complexity. This complexity may pose challenges in practical applications, especially in dynamic or unpredictable environments.
2. The DRL approach, though effective, can be computationally intensive, potentially limiting its use in real-time applications or environments with limited computational resources.
3. The paper does not extensively discuss the scalability of the proposed approach to various robotic systems and tasks. The generalization of this method to different robotic platforms and environments remains unclear.
4. The research primarily focuses on specific test scenarios. Broader testing across diverse applications would strengthen the validity of the proposed method.
Recommendations:
1. Testing in real-world manufacturing environments with varying complexities is recommended to validate the robustness and practicality of the Co-RTSATP framework.
2. Investigating strategies to reduce the computational load of the DRL algorithm could make the method more feasible for real-time applications.
3. Conducting studies on the scalability of the approach to different types of robots and tasks will provide insights into its broader applicability.
4. While the state features for the DRL algorithm are well-thought-out, exploring additional or alternative features could further optimize the learning process and efficiency of the solution.
5. To establish the superiority or particular niche of the Co-RTSATP approach, comparative studies with other existing methods should be expanded.
The paper represents a significant step in robotic optimization research, offering a novel perspective on handling RTSP and ROTP concurrently. However, addressing the noted drawbacks and considering the recommendations could further enhance its contribution to the field.
Author Response
Dear reviewer:
We are truly grateful for your recognition and support of our paper's research content. Your thorough review, detailed comments, and insightful suggestions mean a lot to us. After carefully reading your comments, we implemented specific revisions in the original text in accordance with your suggestions. Please refer to the attachments for specific revison information. Thanks for your reviewing again, sincerely.

Reviewer 2 Report
Comments and Suggestions for Authors
This is a very interesting paper addressing task-sequencing and trajectory-formation challenges in robotic control in an integrated manner. The topic is introduced properly, embedded in relevant earlier work, specified in sufficient detail, investigated properly by comparing the performance of the proposed method with other methods, results reported concisely. Conclusions are carefully drawn and realistically. The major and critical drawback of this ms is its language use. A thorough language revision is required before the suitability for publication can be assessed. In its current state this ms does not meet the criteria for publication. The abovementioned positive and negative feedback have been specified in multiple annotations that I added anonymously to the attached PDF.

Comments on the Quality of English LanguageSee my annotations added to the PDF.
Author Response

(The authors gave the same response as above.)

Reviewer 3 Report
Comments and Suggestions for Authors
The article proposes a novel approach in robotic cell functioning optimization algorithm that combines the problems of optimal path planning and optimal task sequencing in one solution. The proposed approach is well-based and demonstrates good comparative results. Proposed solution has obvious practical value and has a potential to be scaled on multi-robot production units and inclution the consideration of occasionally occuring obstacles between task points which are important issues in practical implementation. Overall, paper is good for publication after minor proofreading.
Comments on the Quality of English LanguageMinor proofreading is required for some misspelling and non-scientific slang over the text. Some examples:
...constantly being restricted by the of trajectory planning results [line 110]
a hot issue of research [line 116]
...and whale algorithm [23] et al. [line 122]
In the Co-RTSATP problem, the the joints’ [line 252]
To verify the superiority of the Co-RTSATP model [line 413]
Author Response
Dear reviewer:
we sincerely thank you for your rigorous review of our manuscript and for your agreement with the views on our research. Your comments are all valuable and very helpful for revising and improving our paper. We implemented specific revisions in the original text in accordance with your suggestions. Please see the attachment for detailed revison information. Thanks again for your comments, sincerely.

Round 2
Reviewer 1 Report
Comments and Suggestions for Authors
Thanks for the authors for considering reviewers' comments and recommendations. In my opinion, now the paper can be accepted
Reviewer 2 Report
Comments and Suggestions for Authors
The authors have handled my comments and suggestions adequately. The ms now meets the standards for publication and is of high quality as specified in my first review.